# DVDeconv: An Open-Source MATLAB Toolbox for Depth-Variant Asymmetric Deconvolution of Fluorescence Micrographs

**DOI:** 10.3390/cells10020397

**Published:** 2021-02-15

**Authors:** Boyoung Kim

**Affiliations:** Robot R&D Group, Factory Automation Technology Team, Global Technology Center, Samsung Electronics, 129, Samsung-ro, Yeongtong, Suwon 443-742, Gyeonggi, Korea; by1110.kim@samsung.com

**Keywords:** 3D microscopy, fluorescence microscopy, deconvolution, blind deconvolution

## Abstract

To investigate the cellular structure, biomedical researchers often obtain three-dimensional images by combining two-dimensional images taken along the z axis. However, these images are blurry in all directions due to diffraction limitations. This blur becomes more severe when focusing further inside the specimen as photons in deeper focus must traverse a longer distance within the specimen. This type of blur is called depth-variance. Moreover, due to lens imperfection, the blur has asymmetric shape. Most deconvolution solutions for removing blur assume depth-invariant or x-y symmetric blur, and presently, there is no open-source for depth-variant asymmetric deconvolution. In addition, existing datasets for deconvolution microscopy also assume invariant or x-y symmetric blur, which are insufficient to reflect actual imaging conditions. DVDeconv, that is a set of MATLAB functions with a user-friendly graphical interface, has been developed to address depth-variant asymmetric blur. DVDeconv includes dataset, depth-variant asymmetric point spread function generator, and deconvolution algorithms. Experimental results using DVDeconv reveal that depth-variant asymmetric deconvolution using DVDeconv removes blurs accurately. Furthermore, the dataset in DVDeconv constructed can be used to evaluate the performance of microscopy deconvolution to be developed in the future.

## 1. Introduction

One of the most basic imaging techniques in biomedical research is wide-field fluorescence microscopy. In wide-field fluorescence microscopy, a dye-labeled specimen is illuminated with light that matches the excitation spectrum of the dye, and emitted light is captured by a camera [1]. Because fluorescent molecules act like light sources located at specific regions within the specimen, the target of interest within the specimen can be observed with high contrast. Additionally, researchers can obtain a three-dimensional (3D) specimen image by taking two-dimensional (2D) fluorescence microscopy images along the z axis [2].

A disadvantage of obtaining 3D specimen images using this technique is that the image captured is blurry because of diffraction and lens aberrations. The blurry image can be modeled by the summation of the product of a clear image and the point spread function (PSF) of the imaging system. The PSF describes the response of the imaging system to a point object. The PSF has a spread shape along the x-y plane and the z axis because out-of-focus intensities enter the in-focus plane. Therefore, the PSF for wide-field microscopy has a unique shape. The PSF in the lateral (x-y) plane contains the Airy disk, while the PSF in the z (x-z or y-z) plane has an hourglass shape. Additionally, some refractive index changes or lens aberration are also present, and the PSF shape is distorted as an asymmetric Airy disk and hourglass shape.

There are two approaches to remove blur in an image. The first approach is to change the hardware to modify the PSF such that it has a more point-like shape. Confocal microscopy and super-resolution microscopy, such as structured illumination microscopy (SIM), photo-activated localization microscopy (PALM), stochastic optical reconstruction microscopy (STORM), and stimulated emission depletion (STED), are included in this approach. Without any post-processing, these microscopy techniques allow researchers to obtain high-quality images. Especially, SIM is suitable for live cell imaging because it is based on conventional microscopy imaging and does not require mechanical movement in the z plane or special fluorescent dyes [3]. However, the first approach has several disadvantages, such as the need for an additional optical setup and a high cost [4]. The second approach for blur removal is the use of a deconvolution algorithm that removes blurs using PSF information. While the second approach requires post-processing, it enables resolution enhancement, contrast enhancement, and denoising without hardware or additional cost [5]. This paper focuses on this second approach. Both approaches for 3D microscopy take time to capture z planes with mechanical movement along z axis. If the user considers live cell imaging, SIM and wide-field microscopy with deconvolution will be suitable because of their fast image acquisition speed.

A high-performance deconvolution algorithm should accurately reflect the imaging environment. Various deconvolution software packages that remove blur in fluorescence micrographs have been released: Huygens, AutoQuant, COSMOS, and DeconvolutionLab [6,7,8,9]. Huygens provides depth-invariant deconvolution processing of large data in parallel. Blind deconvolution and high-speed depth-invariant deconvolution are available in AutoQuant and Deconvolution Lab provides an open-source depth-invariant deconvolution algorithm based on ImageJ. However, it has been reported that the PSF is variant along the z axis [7,10,11,12,13,14,15,16,17] and deconvolution results that do not reflect depth-variance in the PSF have elongated results [18]. COSMOS software provides both depth-invariant and depth-variant deconvolution algorithms using x-y symmetric PSFs [7]. However, even if a deconvolution reflects depth-variance of PSF, if the asymmetry of PSF is not reflected, it causes incorrect results [19]. Based on a review of publications to date, there is no deconvolution algorithm software or toolbox that reflects both the depth-variance and asymmetry of the PSF. In addition, existing synthetic datasets for evaluation assume depth-invariance or x-y symmetry. However, actual images have depth-variant asymmetric blurs. 

This study provides the specifics of depth-invariant and variant deconvolution algorithms using asymmetric PSFs, as well as parameter settings for both PSF generation and deconvolution. An open-source toolbox DVDeconv has been developed to allow users to easily utilize the deconvolution algorithm. To reflect the actual imaging conditions and accurate evaluation, a new dataset is also released and shared in DVDeconv. DVDeconv can be downloaded from github (https://github.com/bykimpage/DVDeconv). Comparisons of deconvolution results of depth-invariant and depth-variant one-step late (OSL) and generalized expectation-maximization (GEM) algorithms used in DVDeconv are also discussed in this paper.

## 2. Materials and Methods

### 2.1. Image Model

An observed 3D image, g, can be modeled by the following summation of the product of the PSF, h, and an object that we want to retrieve, f, under Poisson noise model [13]:(1)$$\;g\left(p_{i}\right)=\mathrm{Poisson}\left(\sum f\left(p_{o}\right)h\left(p_{i},p_{o}\right)\right)=\mathrm{Poisson}\left(f⊗h\right)$$
where $p_{i}=\left\{x_{i},y_{i},z_{i}\right\}$ and $p_{o}=\left\{x_{o},y_{o},z_{o}\right\}$ indicate 3D locations in image space and object space, respectively. In this paper, the sum of products is defined as ⊗. Photons in fluorescence microscopy are under a Poisson distribution because they are collected in a dark room [20]. To easily differentiate them, the negative log-likelihood function of Equation (1) is evaluated as follows:(2)$$J_{data}\left(f\right)=-\log\left(f\right)=\underset{p_{i}}{\sum }\left(f⊗h-g\log\left(f⊗h\right)+\log\left(g!\right)\right)$$

The Poisson image model is weak at noise [16], thus a penalty term is added to the negative log-likelihood function as follows:(3)$$J_{penalized}\left(f\right)=J_{data}+\gamma R\left(f\right)\;where\;R\left(f\right)=\sqrt{\nabla f_{2}^{2}}$$

γ and R denote the regularization parameter, which varies between 0 to 1, and the penalty term, respectively. Ordinarily, the total variation (TV) term is used as the penalty term, which imposes an L2 penalty on differences between adjacent pixels. Image deconvolution with TV restricts noise amplification at simultaneously preserved edges [21]. However, it couples each pixel with its neighbors in such a way that a direct derivative for maximizing the penalized likelihood function is not possible. The most utilized method for maximizing the likelihood function with TV is the OSL. If the image deconvolution method is described as a Richardson–Lucy algorithm with total variation regularization, it typically indicates an OSL algorithm [22]. As the means of maximizing the likelihood function, this method approximates the difference between neighbor pixels of the current image as the difference of the image at the previous iteration. The final form of the OSL algorithm is as follows:(4)$${\hat{f}}^{k+1}=f^{k}\left(h_{mirror}⊗\frac{g}{h⊗f^{k}}\right)\frac{f^{k}}{1-\gamma \mathrm{div}\left(R\left(f^{k}\right)\right)}$$
where k indicates the iteration number and $h_{mirror}=h\left(-p_{o}\right)$ represents the mirrored PSF. As shown in Equation (4), the estimated image at the previous iteration is used for calculating the differentiation of the TV. In other words, OSL approximates the purple surrogate function in Figure 1 and finds the minimum value of the red graph iteratively. Therefore, the OSL can easily obtain the maximum solution of the likelihood function. Unfortunately, the approximated likelihood function does not guarantee a convergence. To overcome this problem, the GEM algorithm is adopted, as suggested in previous work [13,16]. 

The GEM algorithm indirectly evaluates the maximum likelihood using separable quadratic surrogates of the penalty term [23], as described in Figure 1. The final form of the GEM algorithm is as follows [13]:(5)$${\hat{f}}^{\left(k,m+1\right)}=\{\begin{array}{c} \frac{\sqrt{b^{2}\left(f^{\left(k,m\right)}\right)+\overset{ˇ}{c}a\left(f^{\left(k,m\right)}\right)}-b\left(f^{\left(k,m\right)}\right)}{\overset{ˇ}{c}}\;\mathrm{when}\;a\left(f^{\left(k,m\right)}\right)<0\\ \frac{a\left(f^{\left(k,m\right)}\right)}{\sqrt{b^{2}\left(f^{\left(k,m\right)}\right)+\overset{ˇ}{c}a\left(f^{\left(k,m\right)}\right)}+b\left(f^{\left(k,m\right)}\right)}\;\mathrm{when}\;a\left(f^{\left(k,m\right)}\right)\geq 0 \end{array}$$
where $\overset{ˇ}{c}$ and m represent curvature and sub-iteration, respectively. $a\left(f^{\left(k,m\right)}\right)$ and $b\left(f^{\left(k,m\right)}\right)$ are defined as
(6)$$\begin{array}{c} a\left(f^{\left(k,m\right)}\right)=f^{\left(k,m\right)}\left(h_{mirror}⊗\frac{g}{h⊗f^{\left(k,m\right)}}\right)\\ b\left(f^{\left(k,m\right)}\right)=\frac{1}{2}\left(1+\gamma \frac{\partial R\left(f^{\left(k,m\right)}\right)}{\partial f}-\overset{ˇ}{c}f^{\left(k,m\right)}\right) \end{array}$$

The iteration of the GEM algorithm is the process of finding the lowest point of the orange graph, and the sub-iteration is the process of finding the lowest point of the blue graph. Therefore, GEM can find the maximum solution of the likelihood function without approximation of objective function. This iterative technique is slow to converge toward the final result but guarantees the convergence of the cost function. It was previously proven to be effective in biomedical image reconstruction problems [13].

### 2.2. Space-Invariant Deconvolution (Depth-Invariant Deconvolution)

The equations presented thus far can be adapted to a space-variant to invariant image model. Most existing image deconvolution methods in fluorescence microscopy assume space-invariance [24,25,26,27]. If the PSF is invariant, all pixels in the obtained image space have the same blur. The assumption makes f⊗h transform as the convolution f∗h, which is depicted in Figure 2.

Under the space-invariant image model assumption, the final form of the OSL equation can be described as
(7)$${\hat{f}}^{k+1}=f^{k}\left(h_{mirror}*\frac{g}{h*f^{k}}\right)\frac{f^{k}}{1-\gamma \mathrm{div}\left(R\left(f^{k}\right)\right)}$$

$a\left(f^{\left(k,m\right)}\right)$ of the GEM algorithm can also be converted as follows:(8)$$a\left(f^{\left(k,m\right)}\right)=f^{\left(k,m\right)}\left(h_{mirror}*\frac{g}{h*f^{\left(k,m\right)}}\right)$$

As shown in Equations (7) and (8), only one PSF model is needed and the sum of the products of f and h simplifies to one convolution. However, fluorescent micrographs suffer from more severe blur when the microscope focuses deeper within the specimen [10,14]. Therefore, the space-invariant assumption is only effective in a 2D image or for a very shallow specimen.

### 2.3. Depth-Variant Deconvolution

The depth-variant deconvolution implies that blur changes with depth and that the blur is invariant along the same depth. In the depth-variant image model, $f⊗h=\sum f\left(p_{o}\right)h\left(p_{i},p_{o}\right)$ can be expressed as $\sum f\left(p_{o}\right)h\left(p_{i}-p_{o},z_{o}\right)$. This equation shows a sum of convolutions between the specimen plane at a specific depth and the PSF with respect to the depth, as depicted in Figure 3.

As shown in Figure 2 and Figure 3, the observed image depends on the depth-invariance of the PSF. The space-invariant image model can be applicable in the case of a thin specimen, but in most cases, the deconvolution algorithm should reflect the depth-variant image model. The final form of the OSL algorithm reflecting the depth-variant image model is
(9)$${\hat{f}}^{k+1}=f^{k}\left(\sum \frac{g\left(p_{i}\right)h\left(p_{i}-p_{o},z_{o}\right)}{\sum_{p_{o}}f\left(p_{o}\right)h\left(p_{i}-p_{o},z_{o}\right)}\right)\frac{f^{k}}{1-\gamma \mathrm{div}\left(R\left(f^{k}\right)\right)}$$

Additionally, $a\left(f^{\left(k,m\right)}\right)$ of the GEM algorithm under the depth-invariant image model can be converted as
(10)$$a\left(f^{\left(k,m\right)}\right)=f^{\left(k,m\right)}\left(\sum \frac{g\left(p_{i}\right)h\left(p_{i}-p_{o},z_{o}\right)}{\sum_{p_{o}}f^{k}\left(p_{o}\right)h\left(p_{i}-p_{o},z_{o}\right)}\right)$$

Depth-variant image deconvolution algorithms need as many PSF models as the number of z stacks and restore the specimen image using PSFs corresponding to the depth, as shown in Figure 3. The resolution of wide-field fluorescence microscopy is limited by diffraction to about 500 nm along the *z*-axis. Therefore, it is recommended to shoot to cover 500 nm above and below the desired area. In the case of taking a micrograph moving in the *z*-axis every 160 nm, it is recommended to take four more micrographs above and below the region of interest. The required PSFs can be easily generated with the PSF Generator of DVDeconv toolbox, which is handled in detail in the Experimental Setting of Results Section. 

### 2.4. PSF Model

To estimate an accurate specimen image, an accurate PSF acquisition is necessary. For depth-variant deconvolution, PSFs for each depth are required. DVDeconv provides a PSF generator for this purpose. This section describes the PSF model and the method for setting its parameters.

The PSF model in DVDeconv is based on the simplified Zernike polynomial PSF model [19,28]. The Zernike polynomial PSF model is a parametric PSF that includes all aberrations and is expressed as a squared magnitude of the complex-valued amplitude PSF at the emission wavelength:(11)$$h\left(p_{i},z_{o}\right)={\left|h_{A}\left(p_{i},z_{o};\lambda =\lambda_{em}\right)\right|}^{2}$$
where $h_{A}\left(p_{i},z_{o};\lambda =\lambda_{em}\right)$ is the complex-valued amplitude PSF at emission wavelength $\lambda_{em}$. A complex-valued amplitude PSF is defined by
(12)$$h_{A}\left(p_{i},z_{o};\lambda =\lambda_{em}\right)=\iint A\left(\theta_{i},M\right)e^{jk_{0}\left(\varphi_{d}\left(\theta_{i},z_{i}\right)+\varphi_{sp}\left(\theta_{i},\theta_{s}\right)\right)}e^{j\left(k_{x}x+k_{y}y\right)}dk_{y}dk_{x}$$
where $\varphi_{d}\left(\theta_{i},z_{i}\right)$ and $\varphi_{sp}\left(\theta_{i},\theta_{s}\right)$ show the defocus term and spherical aberration. Each term can be written as [19,29]
(13)$$\begin{array}{c} \varphi_{d}\left(\theta_{i},z_{i}\right)=n_{i}z_{i}\left(1-\cos\theta_{i}\right)\\ \varphi_{sp}\left(\theta_{i},\theta_{s}\right)=-z_{o}\left(n_{i}\cos\theta_{i}-n_{s}\theta_{s}\right) \end{array}\;\;\mathrm{where}\;\{\begin{array}{c} \theta_{i}=\sin^{-1}\left(\lambda \sqrt{k_{x}^{2}+k_{y}^{2}}/2\pi n_{i}\right)\\ \theta_{s}=\sin^{-1}\left(\lambda \sqrt{k_{x}^{2}+k_{y}^{2}}/2\pi n_{s}\right) \end{array}$$
where $\theta_{i}$ and $\theta_{s}$ are angles in the immersion medium plane and the object plane, respectively. $n_{i}$ and $n_{s}$ are the refractive index of the immersion and specimen, respectively. $A\left(\theta_{i},M\right)$ in Equation (12) is the product of an apodization function and the Zernike polynomials [29,30,31] that can be written as
(14)$$A\left(\theta_{i},M\right)=\begin{array}{c} \{\begin{array}{c} {\left(\cos\theta_{i}\right)}^{-\frac{1}{2}}\omega \left(k_{x},k_{y},M\right),\;\;\;\;if\;\sqrt{k_{x}^{2}+k_{y}^{2}}<\frac{2\pi \mathrm{NA}}{\lambda }\\ 0,\;\;\;\;\;\;\mathrm{otherwise} \end{array}\;\\ where\;\omega \left(k_{x},k_{y},M\right)=\underset{n=8,12}{\sum }M_{n},Z_{n} \end{array}$$
where $M_{n}$ is the modulus at each Zernike polynomial $Z_{n}$. M represents a collection of moduli. Because Equation (12) already includes the influence of the defocus and the spherical aberration, only x-coma $M_{8}$ and y-coma $M_{12}$ aberrations, which strongly influence the PSF, are added to the Zernike polynomials [19]. If a PSF model that reflects other aberration terms is needed, it is easily achieved by substituting the amount of aberration for $M_{n}$ corresponding to each aberration $Z_{n}$.

## 3. Results

### 3.1. Experimental Setting

#### 3.1.1. PSF Generator

The PSF generator in DVDeconv can generate depth-variant PSFs by simply inputting microscope information. When the PSF generator is launched, initial parameter values are already filled, as shown in Figure 4. The unit of wavelength in the PSF generator is nanometers (nm). The unit of x-y and z resolution is microns (μm). Depth-variant PSFs are generated by the number of values inserted in the # of PSFs prompt. The number of PSFs should be the same as the number of z pixels in the image to be restored. Generated PSFs have voxel sizes based on the user inputted x-y-z pixel values. 

Values input in the x-coma and y-coma aberration prompt cause asymmetry of the PSF. The aberration values are obtained by unconstrained nonlinear optimization. The optimization function finds the best aberration values that maximize the probability of being observed as the captured image from the generated PSF with the aberration values. The value for aberrations can be found from maximizing the following equation [13,19]:(15)$$\left({\hat{\varphi }}_{d},{\hat{\varphi }}_{sp},\;{\hat{M}}_{8},{\hat{M}}_{12}\right)=argmin\underset{p_{i}}{\sum }\left(f⊗h\left(p_{i},z_{o}\right)-g\log\left(f⊗h\left(p_{i},z_{o}\right)\right)+\log\left(g!\right)\right)$$

DVDeconv also provides the code and README file for estimating aberration values.

#### 3.1.2. Deconvolution

DVDeconv provides both depth-invariant and depth-variant OSL and GEM algorithms with deconvolution results dependent on parameter settings. This section discusses the meaning and impact of each parameter value.

The regularization parameter γ has a value between 0 and 1. When γ is close to 1, noise is removed, but image details are destroyed. When γ is 0, the deconvolution algorithm becomes the Richardson–Lucy algorithm. A higher value of γ is preferred when noise is severe.

Because all algorithms in DVDeconv are iterative methods, the number of iterations must be included. Too small a number of iterations will not show enough of a reconstructed deconvolution result, and too large a number of iterations will cause noise amplification. DVDeconv provides a save function for every iterative result. If the user runs the deconvolution algorithms after selecting the every iteration save button, deconvolution results are saved as a tif. or mat. format. The user can choose the result with the approximate number of iterations.

Only the GEM algorithms include the curvature, $\overset{ˇ}{c}$, parameter due to the presence of the surrogate function. If the value of curvature is small, the iteration speed is fast, and the probability of algorithm convergence is low. On the other hand, if the value of curvature is large, the iteration speed is slow, and the convergence probability is high.

### 3.2. Dataset

All of the algorithms in DVDeconv were evaluated on synthetic data. The results applied to the actual data are revealed in the previous paper, and the results of applying the asymmetric depth-variant function showed superior performance [13,19]. At present, there are few open datasets for deconvolution algorithms of micrographs, and those that are available are generated under the assumption of a symmetric PSF [5,7,8], which reflects the actual microscope environment inaccurately. Therefore, a synthetic dataset was generated using a depth-variant asymmetric PSF, the generation source of this dataset is also included in DVDeconv. A blurred object was generated by the summation of the product of an object and 3D depth-variant PSFs at each depth. Then, Gaussian noise was added and the final synthetic image was generated under a Poisson distribution. DVDeconv provides two noise conditions, 10 dB and 15 dB Gaussian noise cases, which are shown in Figure 5.

The generated image has 256 × 256 × 128 voxels of size 64.5 × 64.5 × 160 nm. Dataset images have a dynamic range of 0 to 65535 (uint16). The hollow bars in the synthesized data have a length of 85 pixels. The square of the inner radius of the bars is 15, and that of the outer radius is 43. As shown in Figure 5b, bars that should look the same look differently spread out because different depths have different PSFs. As can be seen from Figure 5c,d, 10 dB image has more noise than the 15dB image. 

### 3.3. Deconvolution Results

Peak signal-to-noise ratio (PSNR), signal-to-noise ratio (SNR), standard deviation of peaks, relative contrast, memory, and processing time were used to measure the performance of the DVDeconv algorithms on synthetic data.

Figure 6 shows deconvolution results from the image in Figure 5c. Images in Figure 6 show the cross-sections cut off the middle of each axis. The value of the regularization parameter γ was set to 0.00001 in the 15dB dataset. As shown in Figure 6a–d, there are still blurs around the bars in x-y section because the depth-invariant deconvolutions only reflect the blur information at a certain depth. In this experiment, depth-invariant algorithms utilized the PSF at the central depth (the 64th pixel). On the other hand, the depth-variant algorithms remove blurs well, as shown in Figure 6e–h. There is almost no difference between the OSL and GEM algorithms in Figure 6.

Figure 7 shows deconvolution results from the image in Figure 5d. The same as Figure 6, images in Figure 7 show the cross sections cut off the middle of each axis. The γ value was set to 0.0006 for the depth-variant GEM algorithm in 10dB dataset. In the rest algorithm, the γ value was set to 0.0001 for the 10dB dataset. Figure 7a–d show more severe blurs around the bars than those in Figure 6a–d because the observed image has more noise. However, despite severe noise, the depth-variant deconvolution algorithms restore bar shapes, as shown in Figure 7e–h. 

There is almost no difference between the OSL and GEM algorithms in Figure 6 and Figure 7. Moreover, there is almost no difference between reconstructed images with the asymmetry applied and those without asymmetry to the naked eye. 

The quantified performance of the deconvolution algorithms was evaluated with PSNR and SNR. PSNR and SNR have been utilized to calculate similarities between the original image and the reconstructed image. A higher value indicates higher image quality. The PSNR and SNR evaluation results are represented in Table 1. Depth-variant algorithms have higher PSNR and SNR values than those of depth-invariant algorithms for both 10 dB and 15 dB images, which is consistent with the qualitative results. The best PSNR and SNR value is shown in bold.

From the results, depth-variant asymmetric algorithms show the best performance. The depth-variant OSL and GEM algorithms have the same PSNR and SNR value for the 15 dB image. For the 10 dB image, the depth-variant asymmetric OSL algorithm shows the best performance. From the results, it can be seen that the performance improves as the characteristics of depth-variance and asymmetry of PSF are applied. PSNR and SNR values of the depth-variant OSL algorithm can have those values of about 0.0258 over depth-variant GEM algorithm. 

Deconvolution algorithms with standard deviation and relative contrasts were also evaluated. Figure 8 and Figure 9 show the intensity profiles of the deconvolution results at different noise levels (15 dB and 10 dB, respectively). The intensity profiles at the center of each bar were normalized by the maximum intensity and plotted as different colors in Figure 8 and Figure 9. The horizontal axes in Figure 8 and Figure 9 designate pixel locations along the *x*-axis. Intensity profiles were normalized by the maximum value of the x-z plane image. Because depth-invariant algorithms restore images using only the PSF at the center of depth, intensity peaks in Figure 8 and Figure 9a–d are uneven compared to those in Figure 8 and Figure 9e–h. This indicates that depth-variant deconvolution is effective regardless of the amount of noise.

To quantify the unevenness of intensity peaks, the standard deviation of peaks (std) and the ratio between the minimum peak and maximum peak of the bars were computed. The closer std value to zero shows the smaller difference between peaks. The closer the relative contrast value to one, the higher the evenness between bars and thus high restoration performance. The std and relative contrasts are shown in Table 2. The best std and relative ratio are shown in bold.

The values of std and relative contrast from depth-variant algorithms show higher intensity evenness in depth than those from depth-invariant algorithms. Different from SNR and PSNR results, the depth-variant asymmetric GEM algorithm performed the best performance. On the other hand, in common with PSNR and SNR results, the deconvolution algorithm improves performance as more PSF characteristics were added. 

We implemented DVDeconv in MATLAB 2016a on Intel CORE i5-6500 processor with Windows 10. Table 3 shows memory requirements and processing time (of one iteration) for each deconvolution algorithm. 

Depth-variant algorithms take more memory and processing time because they estimate the original image using PSFs at every z pixel. With the same depth assumptions, the GEM algorithm spends more processing time than the OSL algorithm because the GEM algorithm also executes sub-iterations for the surrogate function. From the quantified deconvolution results in Table 1 and Table 2, the user would choose depth-variant algorithms for thick specimen images. However, this choice costs approximately thirty-seven times more memory and four to thirty-nine times more processing time. Table 3 and reference [19] provide users with memory requirement and processing time expectations and help users choose an appropriate algorithm.

## 4. Discussion

In this study, a new open-source MATLAB toolbox for deconvolution of fluorescence micrographs, DVDeconv, is investigated. The software provides not only depth-invariant but also depth-variant asymmetric algorithms. Performance of the algorithms was evaluated using SNR, PSNR, std of peaks, relative contrast, memory, and computational time. From experimental results, it is shown that deconvolution algorithms using depth-variant asymmetric PSF remove blurs effectively but require more memory and computational time than depth-invariant algorithms. Moreover, DVDeconv provides a PSF generator and datasets under a realistic assumption of depth-variant asymmetric blur. This work, in conjunction with the DVDeconv toolbox, is expected to assist in research where depth-variant and asymmetric characteristics of blur are applicable, especially in the field of biomedical imaging.

ImageJ has completely outperformed the commercially available microscopy packages in every aspect of image analysis, the deconvolution algorithms remain an unconquered stronghold. If DVDeconv is ported and included in ImageJ, utilizing various deconvolution algorithms will be easier for biologists.

The proposed DVDeconv provides nonlinear deconvolution algorithms. While linear deconvolution does not create higher frequency components above that spatial threshold, nonlinear deconvolution estimates the true image by reviewing the result over multiple iterations. For this reason, it is more effective to enhance an image resolution with nonlinear deconvolution than with linear deconvolution. The nonlinear deconvolution can create the components above the cut-off frequency. As more iterations are executed, nonlinear deconvolution gradually makes the object size small using the observed image and the estimated PSF. However, too many iterations can cause noise amplification and shrinking of objects.

The number of iterations for deconvolution should be set high enough so that convergence is observed. In order to observe the convergence, DVDeconv provides a function to save the deconvolution image after every iteration. Users can find out that is almost no change in the deconvolution images after a certain iteration by checking the saved images. When there is almost no change in the images, it is considered that the algorithm reaches convergence. Users can also observe noise amplification in the images after excessive iteration. All experiments in this paper showed the convergence within 30 iterations. Details on how to reproduce the experimental results are described in README file of DVDeconv.

The regularization parameter of deconvolution closer to 1 reduces noise but destroys image details. The experiments in this paper used the regularization parameter as 0.00001 for 15 dB image. The maximum regularization parameter for 10 dB image was 0.0006. Based on these values, users could adjust the regularization parameter for their image.

For future work, there are xyz variant deconvolution and machine learning approaches that can be applied as blur, there is also a variant along the xy-axis. x-y-z variant deconvolution could help obtain more accurate specimen images, but this algorithm would need more PSFs and more computational resources. In addition, as machine learning algorithms have evolved, machine learning deconvolution for micrographs has been introduced [32,33,34]. Machine learning-based open-source for deconvolution microscopy is expected to be released in the foreseeable future.

## 5. Conclusions

This study established a new open-source MATLAB toolbox called DVDeconv, which provides dataset, PSF generator, and deconvolution algorithms for removing blurs of fluorescence micrographs. DVDeconv reflects actual imaging conditions that blurs are depth-variant and asymmetric. Qualified and quantified deconvolution results verified that the proposed depth-variant asymmetric deconvolution outperforms deconvolutions that do not reflect depth-variance or asymmetry.

DVDeconv takes about 30 min for deconvolution of 256 × 256 × 128 voxels 3D data. The current machine learning algorithm takes 0.4 s for deconvolution of 1024 × 1024 pixels 2D data [33]. It is expected that the machine learning algorithm for 256 × 256 × 128 voxels 3D data would take at least eight times more in terms of the number of voxels. In other words, 3D deconvolution cannot be conducted in real time with both image processing algorithms and machine learning algorithms. However, as GPU performance is advanced, 3D convolution operations in image processing and inferences in machine learning will be accelerated. This will gradually enable 3D image deconvolution in real time.

## Figures and Tables

**Figure 1 cells-10-00397-f001:**
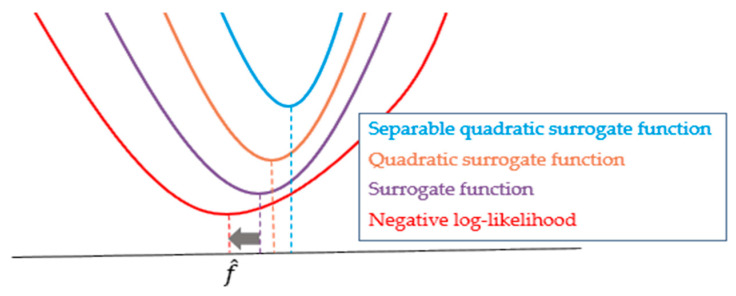
Illustration of the GEM algorithm concept.

**Figure 2 cells-10-00397-f002:**
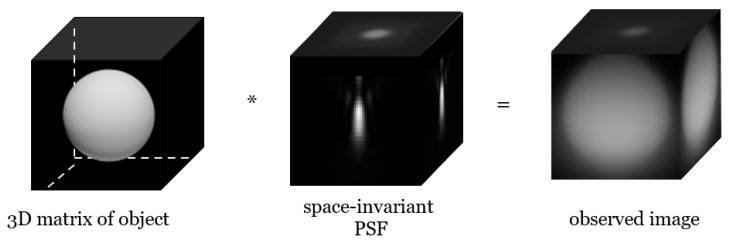
Calculation of the depth-invariant image model.

**Figure 3 cells-10-00397-f003:**
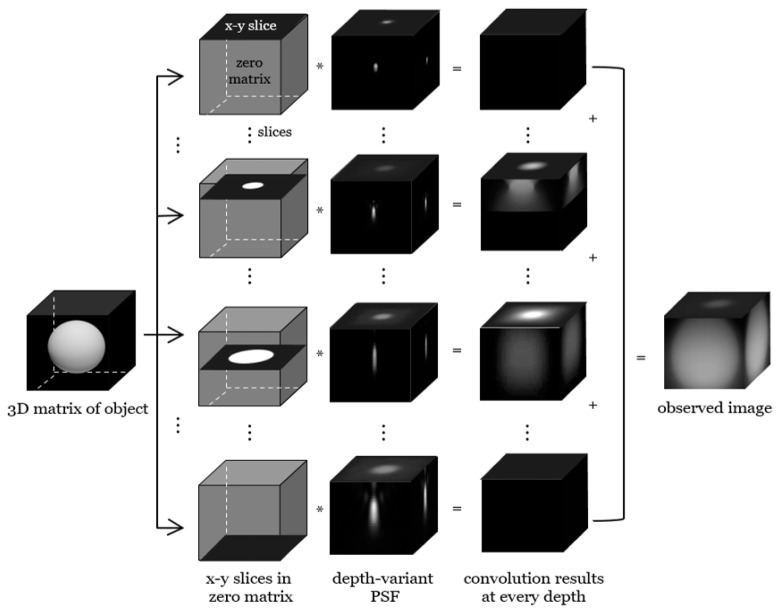
Calculation of the depth-variant image model.

**Figure 4 cells-10-00397-f004:**
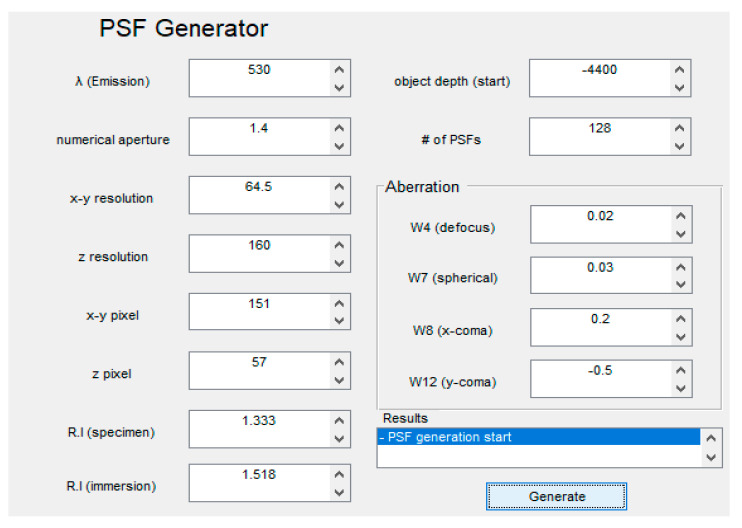
DVDeconv PSF Generator GUI.

**Figure 5 cells-10-00397-f005:**
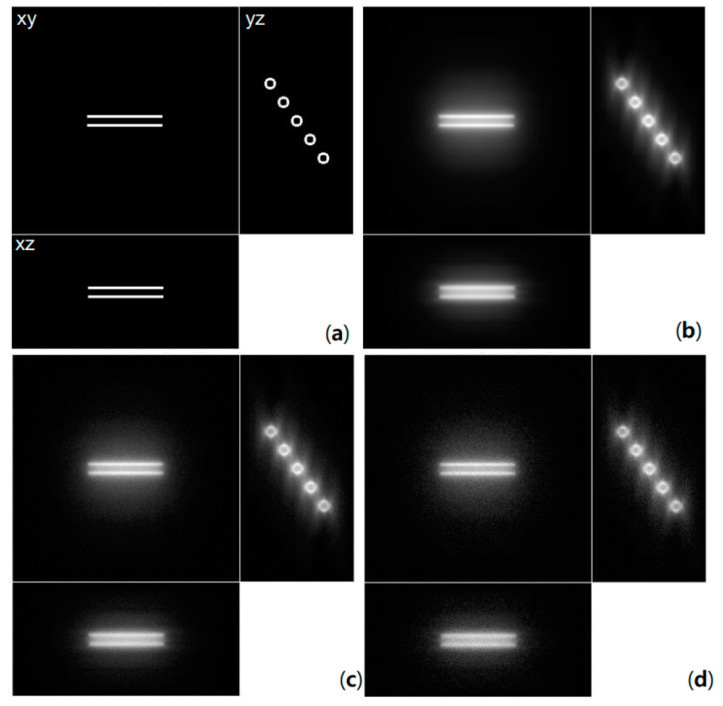
Dataset for DVDeconv evaluation: (**a**) Original image, (**b**) blurred image, and blurred image with (**c**) 15 dB and (**d**) 10 dB Gaussian noise under a Poisson distribution.

**Figure 6 cells-10-00397-f006:**
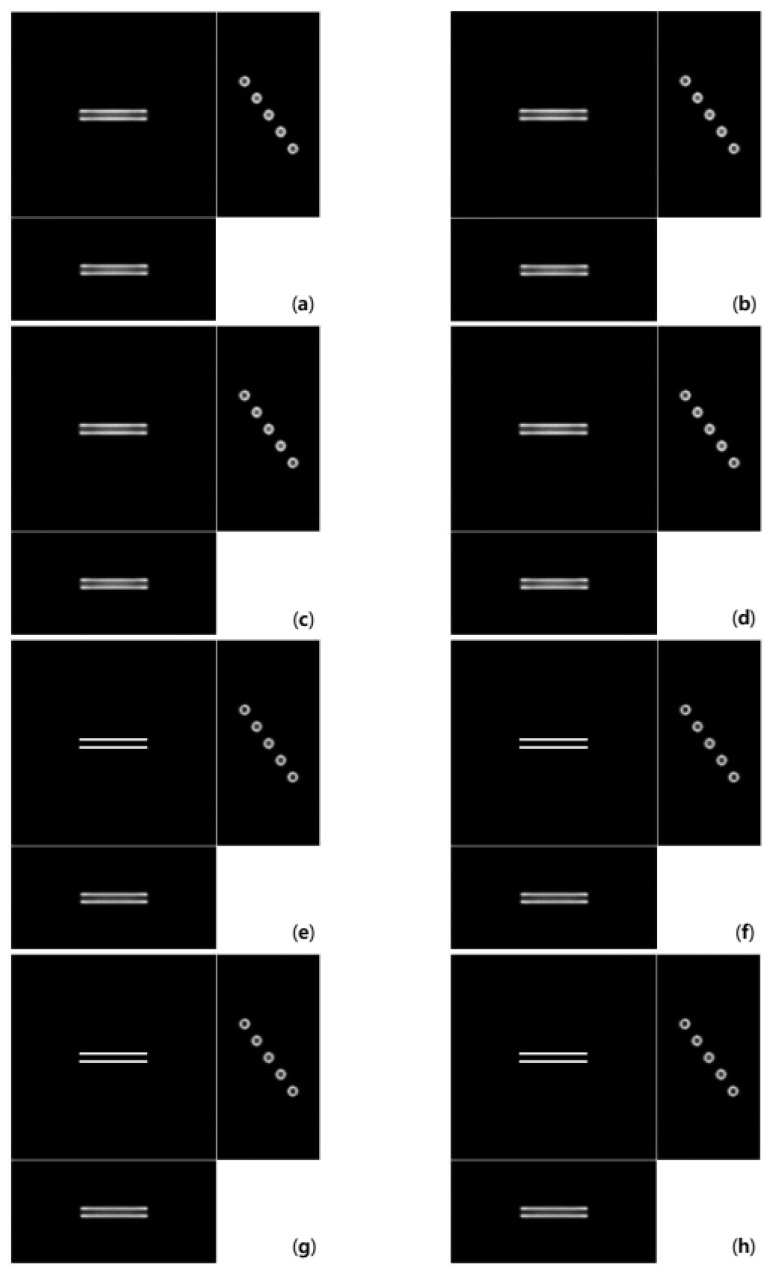
Deconvolution results using the (**a**) depth-invariant symmetric OSL algorithm, (**b**) depth-invariant symmetric GEM algorithm, (**c**) depth-variant symmetric OSL algorithm, (**d**) depth-variant symmetric GEM algorithm (**e**) depth-variant symmetric OSL algorithm, (**f**) depth-variant asymmetric GEM algorithm, (**g**) depth-variant asymmetric OSL algorithm, and (**h**) depth-variant asymmetric GEM algorithm on the 15 dB Gaussian noise image.

**Figure 7 cells-10-00397-f007:**
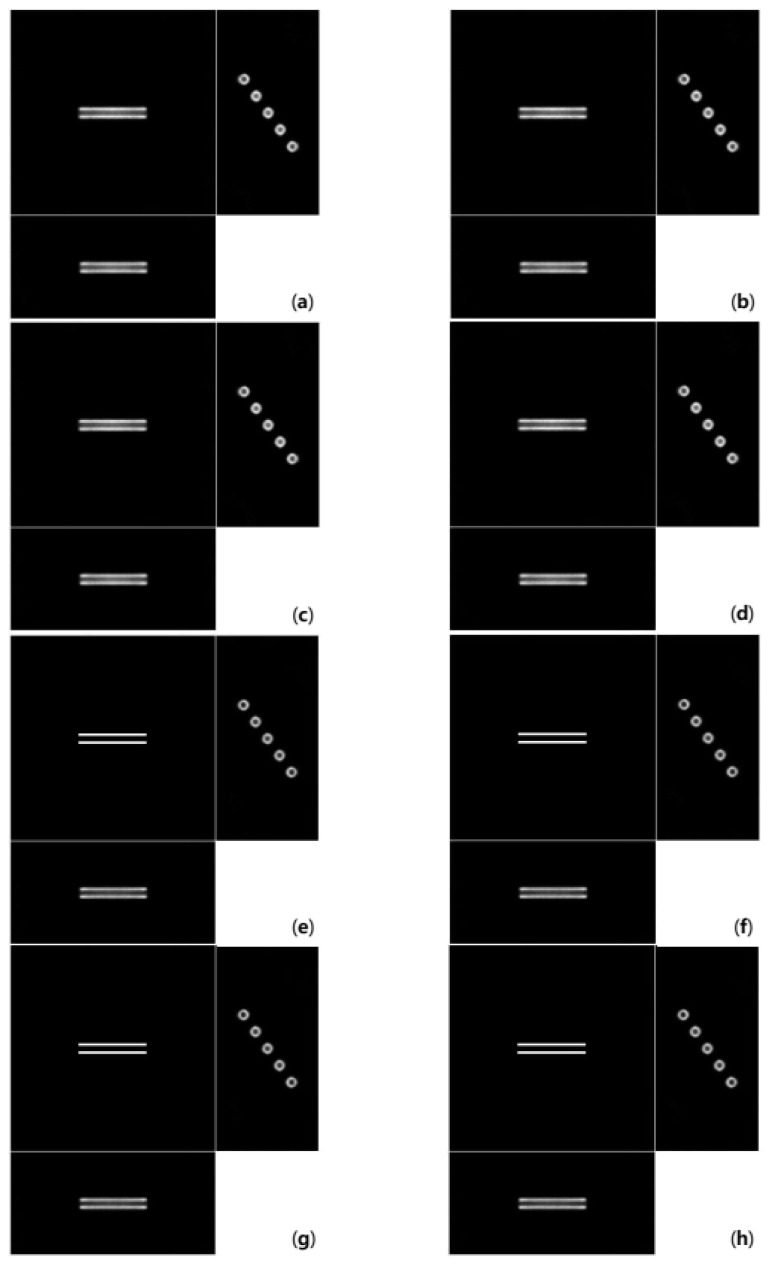
Deconvolution results using the (**a**) depth-invariant symmetric OSL algorithm, (**b**) depth-invariant symmetric GEM algorithm, (**c**) depth-variant symmetric OSL algorithm, (**d**) depth-variant symmetric GEM algorithm, (**e**) depth-variant symmetric OSL algorithm, (**f**) depth-variant asymmetric GEM algorithm, (**g**) depth-variant asymmetric OSL algorithm, and (**h**) depth-variant asymmetric GEM algorithm on the 10 dB Gaussian noise image.

**Figure 8 cells-10-00397-f008:**
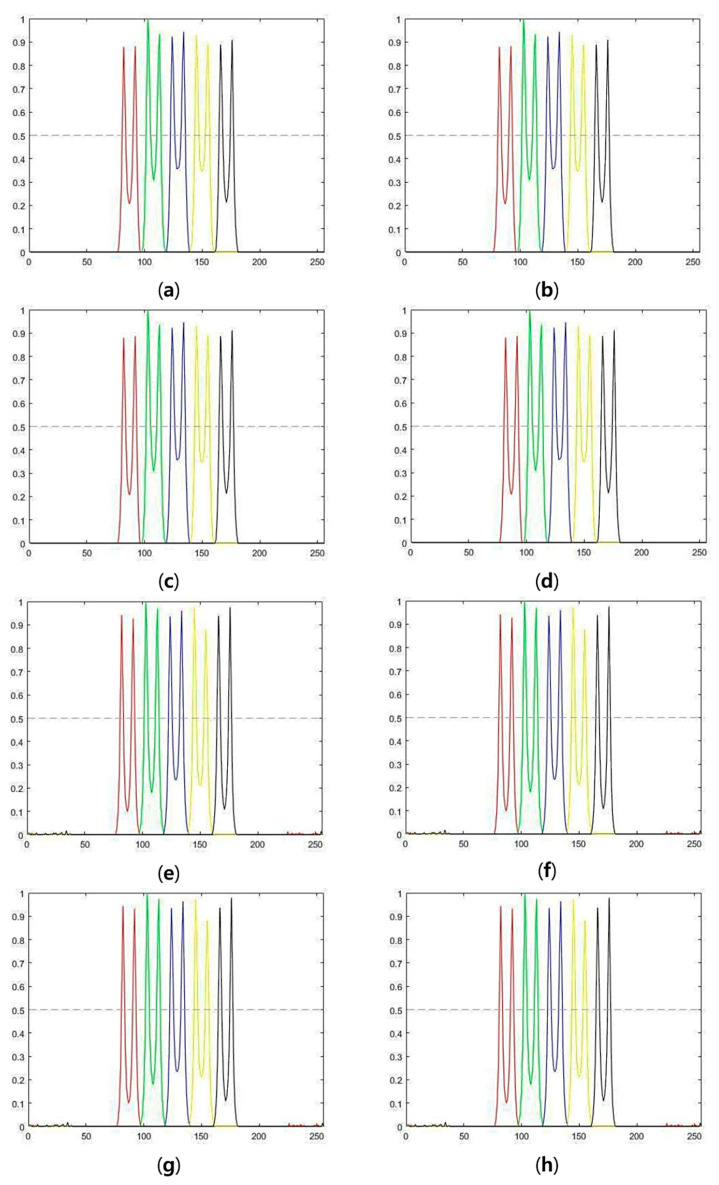
Intensity profiles of deconvolution results for 15 dB images: (**a**) Depth-invariant symmetric OSL algorithm, (**b**) depth-invariant symmetric GEM algorithm, (**c**) depth-variant symmetric OSL algorithm, (**d**) depth-variant symmetric GEM algorithm, (**e**) depth-variant symmetric OSL algorithm, (**f**) depth-variant asymmetric GEM algorithm, (**g**) depth-variant asymmetric OSL algorithm, and (**h**) depth-variant asymmetric GEM algorithm.

**Figure 9 cells-10-00397-f009:**
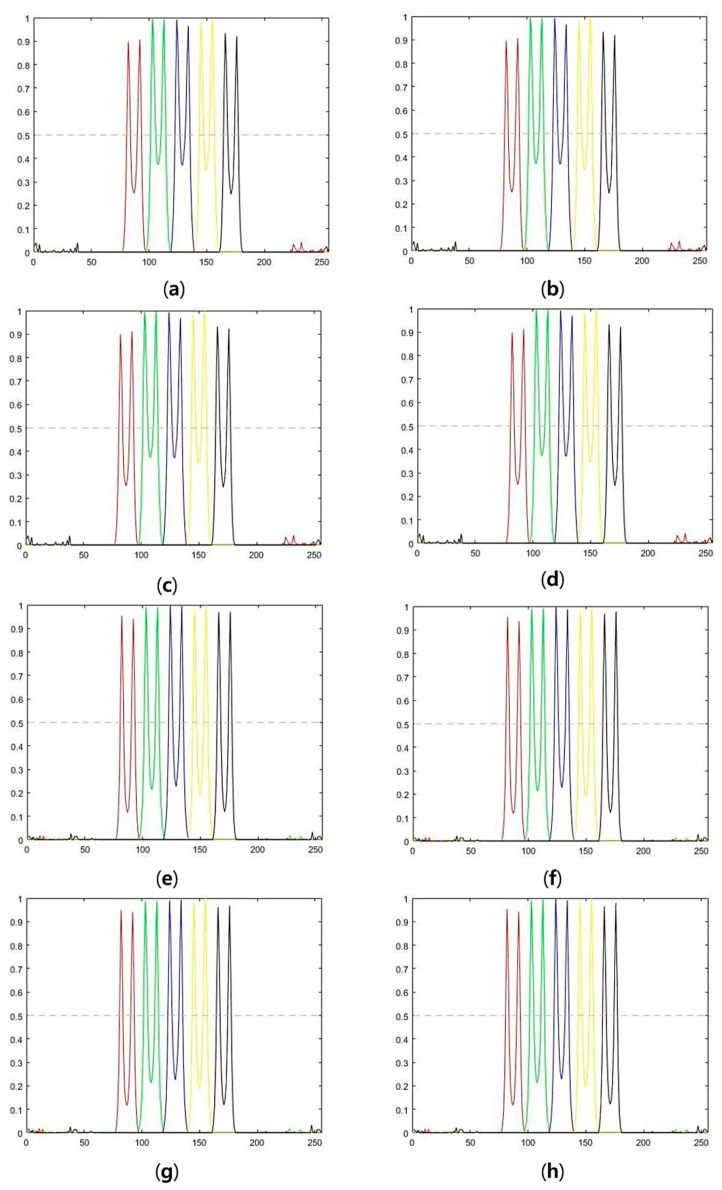
Intensity profiles of deconvolution results for 10 dB images: (**a**) Depth-invariant symmetric OSL algorithm, (**b**) depth-invariant symmetric GEM algorithm, (**c**) depth-variant symmetric OSL algorithm, (**d**) depth-variant symmetric GEM algorithm, (**e**) depth-variant symmetric OSL algorithm, (**f**) depth-variant asymmetric GEM algorithm, (**g**) depth-variant asymmetric OSL algorithm, and (**h**) depth-variant asymmetric GEM algorithm.

**Table 1 cells-10-00397-t001:** Peak signal-to-noise ratio (PSNR) and signal-to-noise ratio (SNR) results.

Depth-Variance	Asymmetry	Surrogate Func (GEM)	PSNR		SNR	
			15 dB	10 dB	15 dB	10 dB
			28.8546	28.6793	5.5394	5.3641
		✓	28.8549	28.6839	5.5398	5.3688
	✓		28.8598	28.6895	5.5447	5.3743
	✓	✓	28.8603	28.6946	5.5452	5.3795
✓			29.1655	28.8552	5.8504	5.5400
✓		✓	29.1655	28.8247	5.8504	5.5096
✓	✓		**29.1933**	**28.8762**	**5.8782**	**5.5611**
✓	✓	✓	**29.1933**	28.8504	**5.8782**	5.5353

**Table 2 cells-10-00397-t002:** Standard deviation of peaks and relative contrast.

Depth-Variance	Asymmetry	Surrogate Func (GEM)	Std		Relative Contrast	
			15 dB	10 dB	15 dB	10 dB
			0.0373	0.0400	0.8780	0.8975
		✓	0.0373	0.0400	0.8780	0.8970
	✓		0.0369	0.0398	0.8788	0.8988
	✓	✓	0.0369	0.0398	0.8788	0.8984
✓			0.0339	0.0203	0.8767	0.9403
✓		✓	0.0339	0.0203	0.8767	0.9359
✓	✓		**0.0332**	0.0201	**0.8812**	0.9404
✓	✓	✓	**0.0332**	**0.0200**	**0.8812**	**0.9414**

**Table 3 cells-10-00397-t003:** Memory requirements and processing time.

	Memory [GB]	Time [s]
Depth-invariant OSL	0.668	1.4
Depth-invariant GEM	0.668	15
Depth-variant OSL	24.5	54.64
Depth-variant GEM	24.5	66.33

## Data Availability

The source code and dataset can be downloaded from github (https://github.com/bykimpage/DVDeconv).

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
