# Peer review of "DVDeconv: An Open-Source MATLAB Toolbox for Depth-Variant Asymmetric Deconvolution of Fluorescence Micrographs"

_cells, 2021, doi:10.3390/cells10020397_

Round 1

Reviewer 1 Report

The submitted paper presents interesting discussion on depth variant deconvolution imaging with a nice open-source toolbox. The paper can be improved with some suggestions below.

Towards the end of paragraph 3 of the introduction (line 55), the author mentions that deconvolution imaging is able to image live cells, however there is no mention that capturing a z-stack is also required, meaning that the speed of capture is limited by the speed of the mechanical movement in the z-plane, along with number of images required in the z-stack, and the acquisition speed of the camera. This also does not necessarily translate to decreased photo-bleaching or even be suitable for live cell imaging. There are other techniques, able to achieve high speed imaging, such as structured illumination microscopy (SIM), which works very well for super resolution live cell work. A reworked discussion with references (such as https://doi.org/10.1046/j.1365-2818.2000.00710.x) including these techniques would improve this section.

Line 76: enables DVDeconv (not, entitled)?

Line 77: To reflect actual imaging (not, For reflecting actual imaging)?

Figure 1: The plots seem to be very low resolution and unreadable, particularly with the orange text. It is also unclear where this figure has come from, was it produced by the author? The figure is also not in references 12 or 15.

Page 5, Paragraph 2: The author mentions that a point spread function (PSF) is needed for each slice of a z-stack, a comment mentioning the minimum number of z-stack slices required for the algorithm would aid the discussion here. In particular, how many slices above and below the sample are required?

Page 6, Results Section: Figure 4 is mostly unreadable and low resolution. A discussion on how to measure the required parameters for the PSF calculator would be very helpful here, particularly for the calculations of the aberrations.

Page 7, Paragraph beginning on Line 227: What bit depth (dynamic range) is being used for the images here?

Figures 8 & 9: Again, low resolution figures, the plots may need layout adjustments to more clearly show the differences between algorithms. It is difficult to see differences between the plots with the current layout.

Section 4, Discussion: A discussion on the non-linearity of deconvolution would be useful here, along with how to choose the number of iterations required for a given signal-to-noise ratio of an image. A comment on over processing or under processing an image would also improve the discussion.

Section 5, Conclusions: A comment on future work, particularly with GPU acceleration, and how it might improve processing times would also be useful.

Reviewer 2 Report

Deconvolution is one of the ever-promising approaches of enhancement of the spatial resolution/removal of the out-of-focus fluorescence. In her work, B Kim presents a Matlab toolbox for 3D deconvolution of the wide-field fluorescence images. The Author is very much correct in identifying a niche for her work: although the likes of ImageJ have completely outperformed the commercially available microscopy packages in every aspect of image analysis, the deconvolution algorithms remain an unconquered stronghold. The PSF variance along the z axis is not a major news though.

I have tested the psf-based 3D deconvolution and can testify that it works in principle and hence should be accepted for publication, provided the Author very carefully reshapes the paper into a more readable/reproducible entity. Whilst the flow of thought is overall traceable, the paper, the formalism (1)-(14) needs additional curation. It looks valid but is poorly explained at the moment, due to the lack of space in the Methods section, I guess. Also confusing is the non-conventional use of mathematical symbols (such as the tilde in eq 1 and the innovative sum of product sign). Statements like "A Poisson distribution with many photons approximates a normal distribution" need to be carefully checked, as in eq 2 the Author substitutes the Poisson by the normal. I am not sure if the term "axial axis" is legitimate. Z is the applicate axis.
